# The Association between *PTPN22* SNPs and susceptibility to type 1 diabetes: An updated meta-analysis

**Yu Su[1], Xue Li[1], Pei-dong Wu[2], Yu-long Zhang[3], Peng-fei Fang[4], Fei-fei Wu[1]*,
Xiao-feng He[5]***

1 Department of Endocrinology, Heping Hospital Affiliated to Changzhi Medical College, Shanxi, Changzhi, China, 2 Shanxi Medical University, Taiyuan, Shanxi, China, 3 Department of Oncology, Handan First Hospital, Handan, Hebei, China, 4 Department of Cardiology, Zhoukou, Henan, China, 5 Institute of Evidence-Based Medicine, Heping Hospital Affiliated to Changzhi Medical College, Changzhi, Shanxi, China

* h4377343@163.com (FFW); 393120823@qq.com (XFH)

## Abstract

Type 1 diabetes (T1D) is a significant global health concern, characterized by the autoimmune destruction of insulin-producing pancreatic β-cells, resulting in lifelong dependence on insulin therapy. Although genetic predisposition plays a crucial role in the pathogenesis of T1D, environmental factors also contribute to its onset and progression. Recent research has identified a number of genetic polymorphisms, particularly in the protein tyrosine phosphatase non-receptor 22 gene (*PTPN22*), that are strongly associated with an increased risk of T1D and may serve as potential biomarkers for early diagnosis and prevention. Despite this, studies investigating the relationship between *PTPN22* rs2476601 and T1D risk have consistently demonstrated an association in certain populations, whereas research on rs1310182 has yielded conflicting and less conclusive results. This study presents an updated meta-analysis of two key *PTPN22* polymorphic loci — rs2476601 (C1858T) and rs1310182 (A852G) — with the aim of clarifying their associations with T1D. The analysis revealed a significant association between *PTPN22* rs2476601 and an increased risk of T1D. In contrast, no significant correlation was found for rs1310182. These findings suggest that *PTPN22* rs2476601 as a marker for T1D susceptibility, offering insights into the development of early intervention strategies. However, further research is required to validate these associations and deepen our understanding of the genetic factors involved in T1D pathogenesis.

## Introduction

T1D is a complex autoimmune disease that results in the destruction of pancreatic β - cells, requiring lifelong insulin therapy [1]. Its pathogenesis is influenced by both genetic predisposition and environmental factors, with an increasing global incidence. In 2017, approximately 230,000 new cases were reported, bringing the total of approximately 9 million cases worldwide. Beyond the HLA region, the *PTPN22* gene has emerged as one of the strongest genetic risk factors for the development of autoimmunity [2].

**Data availability statement:** This meta-analysis utilizes aggregated data from previously published studies. The minimal dataset required to replicate our findings, including extracted effect sizes, heterogeneity statistics, and forest plot data, has been uploaded as Supporting Information Files. No ethical or legal restrictions apply.

**Funding:** The author(s) received no specific funding for this work.

**Competing interests:** There are no conflicts of interest among the authors.

Human genetic polymorphisms significantly affect disease susceptibility, clinical manifestations, and drug response variability. Variants in the *PTPN22* gene have been linked to an elevated risk of various autoimmune diseases, including T1D. *PTPN22* plays a pivotal role in T cell signaling and immune responses, acting as a key desensitization node with several validated and potential substrates [3]. These variations, such as single nucleotide polymorphisms (SNPs), can impact the efficacy and toxicity of chemotherapeutic drugs, leading to inter-individual differences [4]. *PTPN22* encodes lymphatic tyrosine phosphatase (LYP), which negatively regulates T-cell receptor (TCR) signaling. The R620W substitution impairs LYP's interaction with C-terminal Src Kinase (CSK), diminishing its inhibitory effect and enhancing TCR signaling [5]. This alteration in signaling pathways increases susceptibility to autoimmune diseases by affecting T cell tolerance and response. The *PTPN22* rs2476601 polymorphism was first linked to T1D in American and Italian populations by Bottini et al. in 2004 [6], and subsequent studies confirmed this association in Caucasian groups [7–10]. In contrast, the rs2476601 polymorphism was initially reported as non-polymorphic in Asian populations [11]. However, a recent study by Kawasaki et al. suggested that the promoter subregion 1123G/C (rs2488457) of the *PTPN22* gene is associated with acute-onset T1D in the Japanese population, linking this polymorphism to rs2476601 in Caucasians [12]. Some studies propose that the association of 1123G/C (rs2488457) with autoimmune diseases may be due to linkage with C1858T (rs2476601), with the 1858T allele identified as the true susceptibility allele [11].

Similarly, rs1310182 is an intronic sub-polymorphism located within a potential transcription factor binding site of the *PTPN22* gene, It also encompasses the joint-associated protein complex 4 subunit beta 1 (AP4B1) antisense RNA 1 gene [13–16]. Initially, rs1310182 was found to be unrelated to T1D [17], challenging the prevailing view that it is associated with autoimmune disease. However, subsequent studies have reported varying associations, with inconsistency in identifying the specific allele linked to the disease. For example, in Armenians, the rs1310182 T allele was positively associated with T1D, whereas the CC genotype showed a negative association [18]. These associations vary across ethnic groups, as demonstrated in studies from Japan, the UAE, Iran, and Tunisia [14,19–21].

In this context, we selected two SNPs in the *PTPN22* gene, rs2476601 and rs1310182 for this study. Although these polymorphisms have been investigated in 64 studies, the relationship between these SNPs and T1D risk remains unclear due to small sample sizes and conflicting results. Previous meta-analyses have confirmed the association between rs2476601 and T1D risk in Caucasians and American [7–10], but its association in other populations remains uncertain. Additionally, rs1310182 has not been thoroughly explored in relation to clinical characteristics in existing meta-analyses. To address these gaps, we conducted an updated systematic review and meta-analysis, incorporating a more comprehensive data collection process and refining statistical evaluation methods.

## Materials and methods

### Search strategy

A comprehensive literature search was conducted across several databases, including PubMed, Web of Knowledge, China Knowledge, and Wanfang. The search followed PRISMA guidelines and the identified studies were screened initially by title and abstract, followed by a thorough examination of the full texts [22]. Additionally, references from the meta-analyses and reviews were examined to ensure that no relevant studies were overlooked. The search was completed in April 2024. No language or ethnicity restrictions were applied. The search strategy for English-language databases included the following search terms: "polymorphism," "variant," "variation," "mutation," "SNP," "genome-wide association study," "genetic association study,"

"genotype," and "allele," combined with the terms for "type 1 diabetes," "type 1 diabetes mellitus," "type 1 diabetes mellitus," and "diabetes mellitus," and "T1D." Further refinement was made by including the terms "*PTPN22*" and "protein tyrosine phosphatase nonreceptor 22". For the Chinese database: the search terms included "protein tyrosine phosphatase 22," "*PTPN22*," "gene polymorphism," and "type 1 diabetes mellitus."

## Selection criteria

Inclusion criteria: (1) Case-control or cohort studies; (2) Studies examining the correlation between polymorphisms in the *PTPN22* rs2476601 and/or rs1310182 genes and the susceptibility to T1D; (3) Studies reporting genotype frequencies for the genetic polymorphisms in both the case and control groups; (4) Studies conducted on human samples.

Exclusion criteria: (1) Studies with missing or duplicate data; (2) Reviews, letters, and case reports; (3) Studies in which T1D cases are included in the control group; (4) Studies with insufficient or poorly described data.

## Data extraction

To ensure the accuracy of the extracted data, each study meeting the inclusion and exclusion criteria was independently evaluated by two authors. Disagreements were resolved through discussion, and if unresolved, the corresponding author re-extracted and validated the data. For studies with insufficient details, efforts were made to contact the original authors for clarification. Studies with incomplete data were excluded, retaining only high-quality studies and eliminating duplicates. Key information extracted included: the first author's last name, publication year, country, ethnicity (Asian, Caucasian, African, Indian, or mixed), sample size, control source (hospital-based or population-based), type of matching, blinding procedures, quality control in genotyping, and statistical adjustments for assessing associations genotype-T1D associations. These details are summarized in Tables 1 and 2.

## Quality assessment

A quality assessment scale was developed based on the PRISMA guidelines, quality reporting standards for observational studies, and previous meta-analyses [7–10]. Two investigators independently extracted and cross-checked the data, resolving any discrepancies through discussion. If consensus was not reached, a third author conducted a final review. Original authors were contacted for clarification when necessary. S1–S2 Table summarize the quality indicators used to assess the studies. The control group was evaluated using a chi-square goodness-of-fit test, and Hardy-Weinberg equilibrium (HWE) tests were applied to studies with complete genotypic data. A p-value of less than 0.05 indicated significant bias. Studies scoring 12 or higher were classified as high quality, provided they met both the quality scores and HWE criteria. Discrepancies in scores were reviewed by leading experts in the field.

## Statistical analysis

The pooled odds ratios (ORs) and corresponding 95% confidence intervals (CIs) were calculated to evaluate the association between *PTPN22* polymorphisms and the risk of T1D. A p-value of less than 0.05 was considered statistically significant. The *PTPN22* rs2476601 models included: (1) allele model C vs T; (2) additive model CC vs TT; (3) dominant model CC + CT vs TT; (4) recessive model CC vs CT + TT; and (5) super-dominant model CT vs CC. The *PTPN22* rs1310182 models included: (1) allele model C vs T; (2) additive model CC vs TT; (3) dominant model CC + CT vs TT; (4) recessive model CC vs CT + TT; and (5) super-dominant model CT vs CC. HWE was assessed using the goodness-of-fit test, with significant deviation

**Table 1. Characteristics of studies on rs2476601 included in the meta-analysis.**

| First author/ Year | Country | Ethnicity | SC | Type of control | Matching | Sample size | Genotypes distribution of *PTPN22*(rs2476601) | | | | | | HWE | Quality score |
|---|---|---|---|---|---|---|---|---|---|---|---|---|---|---|
| | | | | | | | Cases | | | Controls | | | | |
| | | | | | | | CC | CT | TT | CC | CT | TT | | |
| Smyth et al.2004 [23] | U.K. | Caucasian | PB | Normoglyce-mic controls | NA | 294/395 | 661 | 222 | 18 | 669 | 165 | 10 | 0.9613 | 13 |
| Bottini et al.2004 [6] | U.S | Caucasian | HB | Healthy controls | Age and sex | 294/395 | 193 | 90 | 11 | 307 | 84 | 4 | 0.5070 | 14 |
| Bottini et al.2004 [6] | Italian | Caucasian | HB | Healthy controls | Age and sex | 174/214 | 158 | 15 | 1 | 205 | 9 | 0 | 0.7534 | 13 |
| Zheng et al. 2005 [24] | U.S | Caucasian | PB | Normoglyce-mic controls | NA | 396/1178 | 290 | 97 | 9 | 984 | 186 | 8 | 0.8043 | 16 |
| Kahles et al. 2005 [25] | German | Caucasian | PB | Healthy controls | Sex | 220/239 | 142 | 71 | 7 | 187 | 50 | 2 | 0.4978 | 13 |
| Gomez et al. 2005 [26] | Colombian | Mixed | PB | Normoglyce-mic controls | Sex | 110/308 | 94 | 38 | 1 | 281 | 27 | 0 | 0.4211 | 10 |
| Zhernakova et al. 2005 [27] | Netherlands | Caucasian | PB | NR | Sex | 334/528 | 226 | 96 | 12 | 440 | 84 | 4 | 0.9967 | 11 |
| Hermann et al. 2006 [28] | Finland | Caucasian | HB | Healthy controls | Age and sex | 546/538 | 316 | 200 | 30 | 402 | 122 | 14 | 0.2027 | 16 |
| Fedetz et al.2006 [29] | Ukraine | Caucasian | HB | Normoglyce-mic controls | Age and sex | 296/242 | 187 | 93 | 16 | 176 | 64 | 2 | 0.1394 | 14 |
| Steck et al.2006 [30] | U.S | Caucasian | HB | Normoglyce-mic controls | Age | 690/515 | 482 | 193 | 15 | 425 | 87 | 3 | 0.5202 | 15 |
| Chelala et al.2006 [31] | France | Mixed | PB | NR | Sex | 885/442 | 623 | 243 | 19 | 442 | 363 | 73 | 0.8995 | 12 |
| Santiago et al.2007 [32] | Spain | Caucasian | PB | Healthy controls | Age and sex | 554/316 | 483 | 68 | 3 | 252 | 59 | 5 | 0.4756 | 17 |
| Nielsen et al.2007 [33] | Danish | Caucasian | PB | Healthy controls | Age and sex | 253/354 | 182 | 61 | 10 | 289 | 65 | 0 | 0.0572 | 13 |
| Cinek et al.2007 [34] | Czech | Caucasian | PB | Normoglyce-mic controls | Age and sex | 372/400 | 231 | 127 | 14 | 323 | 72 | 5 | 0.6647 | 16 |
| Cinek et al.2007 [34] | Azeri | Caucasian | PB | Normoglyce-mic controls | Age and sex | 160/271 | 152 | 7 | 1 | 269 | 2 | 0 | 0.9514 | 15 |
| Baniasadi et al.2008 [35] | India | Asian | PB | Healthy controls | Age and sex | 129/109 | 121 | 7 | 1 | 103 | 6 | 0 | 0.7676 | 14 |
| Douroudis et al.2008 [36] | Estonia | Caucasian | HB | Healthy controls | Age and sex | 170/230 | 99 | 57 | 14 | 172 | 52 | 6 | 0.3941 | 15 |
| Dultz et al.2008 [37] | German | Caucasian | PB | Healthy controls | Age and sex | 70/100 | 55 | 15 | 0 | 86 | 12 | 2 | 0.0646 | 13 |
| Smyth et al.2008 [38] | U.K. | Caucasian | PB | Healthy controls | Age | 8984/10930 | 5030 | 2165 | 239 | 5762 | 1230 | 61 | 0.6024 | 11 |
| Korolija et al.2009 [39] | Croatia | Caucasian | PB | Healthy controls | Age and sex | 102/193 | 47 | 51 | 4 | 149 | 43 | 1 | 0.2566 | 10 |
| Lavrikova et al.2009 [40] | Russia | Caucasian | PB | Healthy controls | Age and sex | 162/203 | 108 | 49 | 5 | 159 | 41 | 3 | 0.848 | 12 |
| Fichna et al.2010 [41] | Polish | Caucasian | PB | Normoglyce-mic controls | Age and sex | 215/236 | 143 | 64 | 8 | 185 | 47 | 4 | 0.6149 | 13 |
| Kordonouri et al.2010 [42] | German | Caucasian | PB | Healthy controls | Age and sex | 243/209 | 177 | 58 | 8 | 167 | 40 | 2 | 0.8166 | 14 |
| Chagastelles et al.2010 [43] | Brazilian | Caucasian | HB | Healthy controls | Age and sex | 211/241 | 152 | 56 | 3 | 216 | 25 | 0 | 0.3957 | 14 |
| Zhebrun et al.2011 [44] | Russia | Caucasian | PB | Healthy controls | Age and sex | 150/200 | 99 | 40 | 11 | 132 | 66 | 2 | 0.0434 (HWD) | 12 |

*(Continued)*

**Table 1.** (Continued)

| First author/Year | Country | Ethnicity | SC | Type of control | Matching | Sample size | Genotypes distribution of *PTPN22*(rs2476601) | | | | | | HWE | Quality score |
|---|---|---|---|---|---|---|---|---|---|---|---|---|---|---|
| | | | | | | | Cases | | | Controls | | | | |
| | | | | | | | CC | CT | TT | CC | CT | TT | | |
| Liu et al.2012 [45] | China | Asian | PB | Healthy controls | Age and sex | 229/210 | 228 | 1 | 0 | 210 | 0 | 0 | 0.0000 (HWD) | 12 |
| Kisand et al.2012 [46] | Estonia | Caucasian | HB | Healthy controls | Age and sex | 154/229 | 88 | 53 | 13 | 172 | 51 | 6 | 0.3531 | 14 |
| Giza et al.2013 [47] | Greek | Caucasian | PB | Normoglycemic controls | Age and sex | 130/135 | 116 | 13 | 1 | 127 | 8 | 0 | 0.7228 | 15 |
| Hadzija et al.2013 [48] | Bosnia and Herzegovina | Caucasian | PB | Normoglycemic controls | Age and sex | 241/161 | 128 | 100 | 13 | 102 | 57 | 2 | 0.0525 | 15 |
| Almasi et al.2014 [49] | Iran | Caucasian | PB | Healthy controls | Age and sex | 144/197 | 140 | 4 | 0 | 191 | 6 | 0 | 0.8282 | 12 |
| Kumar et al.2014 [50] | India | Indian | HB | Healthy controls | Age and sex | 145/210 | 137 | 8 | 0 | 208 | 2 | 0 | 0.9447 | 12 |
| Min et al.2014 [51] | China | Asian | PB | Healthy controls | Age and sex | 19/20 | 13 | 4 | 2 | 16 | 4 | 0 | 0.6193 | 13 |
| Liu et al.2015 [52] | China | Asian | HB | Healthy controls | Age and sex | 239/213 | 195 | 38 | 6 | 197 | 15 | 1 | 0.2373 | 13 |
| Pawlowicz et al.2017 [53] | Poland | Mixed | PB | Healthy controls | Age and sex | 147/327 | 91 | 44 | 12 | 199 | 118 | 10 | 0.1299 | 14 |
| Heneberg et al.2018 [13] | Czech | Caucasian | HB | Healthy controls | Age and sex | 263/400 | 168 | 84 | 11 | 323 | 72 | 5 | 0.6647 | 15 |
| Alswat et al.2018 [54] | Saudi | Asian | HB | Normoglycemic controls | Age and sex | 372/372 | 290 | 66 | 16 | 348 | 18 | 6 | 0.0000 (HWD) | 14 |
| El Fotoh et al.2019 [55] | Egyptia | Mixed | HB | Normoglycemic controls | Age and sex | 120/120 | 81 | 33 | 6 | 108 | 12 | 0 | 0.5642 | 17 |
| Rochmah et al.2023 [56] | Indonesia | Mixed | HB | Healthy controls | Age and sex | 31/31 | 3 | 28 | 0 | 0 | 31 | 0 | 0.0000 (HWD) | 13 |
| Zak et al.2023 [18] | Armenia | Caucasian | HB | Healthy controls | Sex | 96/100 | 87 | 9 | 0 | 97 | 3 | 0 | 0.1190 | 14 |

HB = hospital-based studies, PB = population-based studies, HWE= Hardy-Weinberg equilibrium, HWD = Hardy-Weinberg Disequilibrium, NR = Not reported.

**Table 2. Characteristics of studies on rs1310182 included in the meta-analysis.**

| First author/Year | Country | Ethnicity | SC | Type of control | Matching | Sample size | Genotypes distribution of *PTPN22* rs1310182 | | | | | | HWE | Quality score |
|---|---|---|---|---|---|---|---|---|---|---|---|---|---|---|
| | | | | | | | Cases | | | Controls | | | | |
| | | | | | | | CC | CT | TT | CC | CT | TT | | |
| Taniyama et al.2010 [14] | Japan | Asian | PB | Healthy controls | Age and sex | 184/179 | 103 | 66 | 15 | 120 | 51 | 8 | 0.3952 | 12 |
| Sharma et al.2012 [21] | Emirati nationals | Asian | HB | Healthy controls | Age and sex | 139/171 | 24 | 50 | 65 | 54 | 86 | 31 | 0.7499 | 15 |
| Zouidi et al.2014 [20] | Republic of Tunis | Caucasian | HB | Healthy controls | Age | 76/151 | 7 | 40 | 29 | 20 | 66 | 65 | 0.6176 | 15 |
| Abbasi et al.2017 [19] | Iran | Caucasian | PB | Healthy controls | Age | 99/100 | 26 | 40 | 33 | 19 | 47 | 34 | 0.7013 | 12 |
| Heneberg et al.2018 [13] | Czech | Caucasian | HB | Healthy controls | Age and sex | 248/100 | 103 | 105 | 40 | 31 | 53 | 16 | 0.3987 | 16 |
| Zak et al.2023 [18] | Armenia | Caucasian | HB | Healthy controls | Sex | 96/100 | 13 | 58 | 25 | 43 | 39 | 18 | 0.0930 | 14 |

HB = hospital-based studies, PB = population-based studies, HWE= Hardy-Weinberg equilibrium.

defined as Hardy-Weinberg Disequilibrium (HWD) when P < 0.05. Subgroup analyses were conducted based on factors such as genotyping quality, HWE status, control source, geographic region, and ethnicity. The chi-square Q-test and $I^2$ statistic assessed heterogeneity, with a p-value greater than 0.10 or $I^2$ less than 50% indicating no significant heterogeneity, prompting the use of a fixed-effects model [57]. Otherwise, a random-effects model was applied [58]. In cases of significant heterogeneity, meta-regression analysis was performed to identify the sources of heterogeneity. Robustness was assessed through sensitivity analyses, including the removal of individual studies, exclusion of low-quality studies and those with HWD, and selection of high-quality studies meeting HWE criteria. Begg's funnel plot and Egger's test were used to evaluate the stability of the results [59]. In cases where publication bias was detected [60], the non-parametric 'trim-and-fill' method was used to adjust for and identify asymmetries in the funnel plot due to publication bias, while also estimating the true value of the composite measure [61]. Furthermore, the confidence in significant results was evaluated using the following criteria: false positive reporting probability (FPRP), Bayesian false discovery probability (BFDP) and Venice criteria [62,63]. Significant correlations were considered 'positive results' if they met the following statistical criteria: (1) a p-value < 0.05 in at least two genetic models; (2) FPRP < 0.2 and BFDP < 0.8; (3) statistical efficiency > 0.8; (4) $I^2$ < 50%. All statistical analyses were conducted using STATA 12.0.

## Results

### Description of included studies

A total of 1,581 articles were retrieved using a structured search strategy (see Fig 1 for the search and selection process). After screening titles, abstracts, and full texts, 41 articles and 45 studies met the inclusion criteria for the meta-analysis. These studies, published

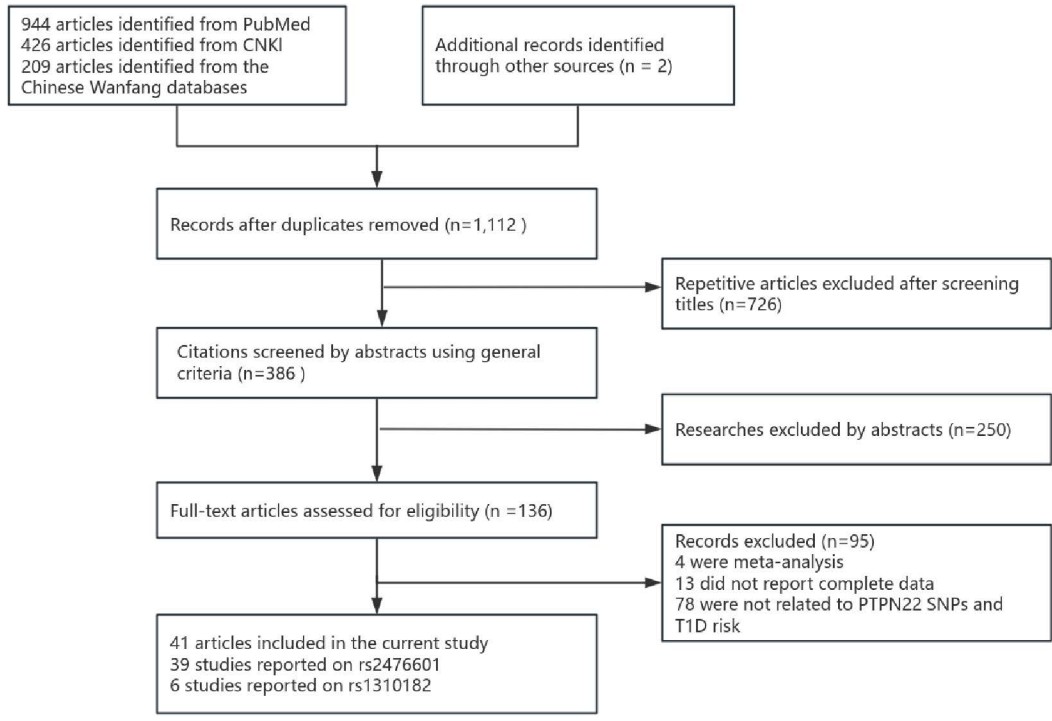

**Fig 1. Flowchart of Literature Retrieval Selection.**

between 2004 and 2023, comprised 19,186 cases of type 1 diabetes and 22,512 controls. Specifically, 39 studies investigated the *PTPN22* rs2476601 polymorphism, while six focused on *PTPN22* rs1310182. Among the included studies, 32 studies were conducted on Caucasian populations, seven on Asian populations, five on mixed-race populations, one on Indian populations, and none on African populations. To minimize the impact of low-quality studies, we identified the quality of the included research, we identified 32 high-quality studies (scoring >12 points) on the rs2476601 polymorphism and four on rs1310182. The genotype distribution of control groups was consistent with HWE in all but two studies. Tables 1–2 present the genotype frequencies and HWE test results of rs2476601 and 1310182 in relation to T1D risk.

## Meta-analysis results

**Pooled analysis for *PTPN22* rs2476601.** This meta-analysis included 39 case-control studies (18,344 cases and 21,711 controls) examining the association between *PTPN22* rs2476601 polymorphism and susceptibility to T1D. As shown in Table 3, significant associations with T1D risk were observed across all genetic models compared to the control group, with the following odds ratios (ORs) and 95% confidence intervals (CIs): CC vs. TT: OR = 0.298, 95% CI = 0.193–0.459; CT vs. CC: OR = 1.777, 95% CI = 1.484–2.128; CT+TT vs. CC: OR = 0.528, 95% CI = 0.437–0.639; CT+CC vs. TT: OR = 0.341, 95% CI = 0.231–0.502; C vs. T: OR = 0.547, 95% CI = 0.426–0.701. In race-based subgroup analyses, the *PTPN22* rs2476601 gene polymorphism was significantly associated with the risk of T1D in Caucasians (CC vs. TT: OR = 0.254, 95% CI = 0.211–0.305; CT vs. CC: OR = 1.764, 95% CI = 1.553–2.004) and Asian races (CC vs. TT: OR = 0.277, 95% CI = 0.123–0.622; CT vs. CC: OR = 2.921, 95% CI = 2.027–2.207). However, these associations were not statistically significant in mixed populations. In population-based studies, negative associations were observed when the data were stratified by control source and type. Collectively, these findings suggest that *PTPN22* polymorphisms may increase the risk of T1D.

**Pooled analysis for *PTPN22* rs1310182.** Table 4 presents a pooled analysis of the *PTPN22* rs1310182 polymorphism and its association with T1D risk, based on six case-control studies involving 842 cases and 801 controls. No statistically significant associations were found in any model (CC vs. TT: OR = 0.560, 95% CI = 0.270–1.162; CT vs. CC: OR = 1.325, 95% CI = 0.740–2.371), with consistent results across subgroup analyses stratified by control source.

## Heterogeneity and sensitivity analyses

Heterogeneity was observed for both SNPs in the overall and subgroup analyses. For rs2476601, significant heterogeneity was detected in the following comparisons, along with their respective $I^2$ and p-values: CC vs. TT ($I^2$ = 77.5%, P < 0.001), CT vs. CC ($I^2$ = 87.1%, P < 0.001), CC vs. CT+TT ($I^2$ = 89.3%, P < 0.001), CT+CC vs. TT ($I^2$ = 74.3%, P < 0.001), and C vs. T ($I^2$ = 95.5%, P < 0.001). These results indicate substantial heterogeneity across all five inheritance patterns. Similarly, for rs1310182, significant heterogeneity was observed in the following comparisons: CC vs. TT ($I^2$ = 80.3%, P < 0.001), CT vs. CC ($I^2$ = 80.5%, P = 0.004), CC vs. CT+TT ($I^2$ = 83.2%, P < 0.001), CT+CC vs. TT ($I^2$ = 93.5%, P < 0.001), and C vs. T ($I^2$ = 85.5%, P < 0.001). Meta-regression analyses were conducted to identify potential sources of heterogeneity, including ethnicity, sample size, control source, case source, HWE, family history, association assessment with appropriate statistics, and adjustment for confounders and quality scores. The results indicated that ethnicity (C vs. T: P = 0.016) and HWE (C vs. T: P = 0.024) were significant sources of heterogeneity for the *PTPN22* rs476601 polymorphism and its association with T1D risk across studies.

**Table 3. Meta-analysis of the association of PTPN22 rs2476601 polymorphism with risk of T1D.**

| Variable | n (Cases/Controls) | CC vs. TT OR (95% CI) | Ph/I2(%) | CT vs. CC OR (95% CI) | Ph/I2(%) | CC vs. CT+TT OR (95% CI) | Ph/I2(%) | CT+CC vs. TT OR (95% CI) | Ph/I2(%) | C vs. T OR (95% CI) | Ph/I2(%) |
|---|---|---|---|---|---|---|---|---|---|---|---|
| **Overall** | 39 (18344/21711) | 0.298 (0.193-0.459) * | 0.000/77.5 | 1.777 (1.484-2.128) * | 0.000/87.1 | 0.528 (0.437-0.639) * | 0.000/89.3 | 0.341 (0.231-0.502) * | 0.000/71.3 | 0.547 (0.426-0.701) * | 0.000/95.5 |
| **Ethnicity** | | | | | | | | | | | |
| Caucasian | 28 (15918/19349) | 0.254 (0.211-0.305) | 0.358/7.2 | 1.764 (1.553-2.004) * | 0.000/68.6 | 0.534 (0.470-0.607) * | 0.000/70.7 | 0.295 (0.246-0.355) | 0.479/0.0 | 0.553 (0.429-0.714) * | 0.000/94.9 |
| Asian | 5 (988/924) | 0.277 (0.123-0.622) | 0.930/0.0 | 2.921 (2.027-4.207) | 0.132/43.4 | 0.356 (0.229-0.554) * | 0.271/22.5 | 0.306 (0.137-0.683) | 0.916/0.0 | 0.333 (0.245-0.454) | 0.485/0.0 |
| Mixed | 5 (1293/1228) | 0.473 (0.055-4.094) * | 0.000/91.8 | 1.249 (0.449-3.473) * | 0.000/94.8 | 0.769 (0.256-2.308) * | 0.000/95.9 | 0.490 (0.068-3.549) * | 0.000/90.2 | 0.686 (0.281-1.675) * | 0.000/96.2 |
| **Geographic region** | | | | | | | | | | | |
| Western Europe | 3 (9612/11853) | 0.277 (0.148-0.518) * | 0.086/59.3 | 1.819 (1.382-2.392) * | 0.005/81.4 | 0.524 (0.388-0.707) * | 0.001/85.4 | 0.281 (0.217-0.363) | 0.137/49.6 | 0.543 (0.408-0.722) * | 0.001/86.7 |
| Southern Europe | 5 (1201/1019) | 0.374 (0.078-1.784) * | 0.028/63.2 | 1.607 (0.786-3.284) * | 0.000/88.1 | 0.592 (0.278-1.264) * | 0.000/89.9 | 0.437 (0.109-1.758) * | 0.069/54.1 | 0.616 (0.311-1.220) * | 0.000/90.0 |
| Northern Europe | 4 (1123/1351) | 0.284 (0.177-0.455) | 0.340/10.5 | 1.950 (1.587-2.287) | 0.575/0.0 | 0.483 (0.405-0.576) | 0.783/0.0 | 0.351 (0.220-0.559) | 0.288/20.4 | 0.505 (0.434-0.588) | 0.941/0.0 |
| Eastern Europe | 3 (608/645) | 0.183 (0.079-0.424) | 0.475/0.0 | 1.252 (0.822-1.907) * | 0.066/63.2 | 0.704 (0.553-0.896) | 0.164/44.7 | 0.194 (0.084-0.447) | 0.391/0.0 | 0.660 (0.535-0.816) | 0.429/0.0 |
| Central Europe | 6 (1383/1584) | 0.300 (0.177-0.508) | 0.709/0.0 | 1.987 (1.673-2.361) | 0.418/0.0 | 0.482 (0.408-0.570) | 0.407/1.4 | 0.356 (0.211-0.602) | 0.739/0.0 | 0.460 (0.112-1.885) * | 0.000/98.8 |
| Asia | 10 (1564/1733) | 0.270 (0.123-0.593) | 0.974/0.0 | 2.746 (2.008-3.755) | 0.067/43.8 | 0.345 (0.257-0.464) | 0.102/38.5 | 0.299 (0.137-0.651) | 0.964/0.0 | 0.588 (0.328-1.054) * | 0.000/71.3 |
| **Source of control** | | | | | | | | | | | |
| PB | 24 (14543/17661) | 0.345 (0.102-0.656) * | 0.000/84.7 | 1.533 (1.190-1.974) * | 0.000/91.0 | 0.614 (0.469-0.804) * | 0.000/92.6 | 0.382 (0.215-0.679) * | 0.000/80.5 | 0.724 (0.549-0.955) * | 0.000/94.6 |
| HB | 15 (3801/4050) | 0.257 (0.183-0.361) | 0.974/0.0 | 2.121 (1.888-2.382) | 0.038/43.3 | 0.427 (0.363-0.502) | 0.052/40.5 | 0.309 (0.221-0.433) | 0.958/0.0 | 0.349 (0.213-0.572) * | 0.000/95.5 |
| **Type of control** | | | | | | | | | | | |
| Healthy controls | 25 (13729/16408) | 0.275 (0.206-0.368) | 0.260/15.6 | 1.694 (1.411-2.034) * | 0.000/73.4 | 0.504 (0.474-0.536) * | 0.000/74.1 | 0.291 (0.237-0.356) | 0.378/6.3 | 0.490 (0.360-0.665) * | 0.000/94.2 |
| Non-diabetic controls | 12 (339/4333) | 0.293 (0.203-0.424) | 0.868/0.0 | 1.675 (1.508-1.860) * | 0.000/71.3 | 0.512 (0.460-0.571) * | 0.000/72.3 | 0.335 (0.232-0.484) | 0.904/0.0 | 0.599 (0.419-0.856) * | 0.000/91.5 |
| **Blinding and/or Quality control** | | | | | | | | | | | |
| YES | 20 (5121/5983) | 0.271 (0.202-0.363) | 0.381/6.3 | 1.904 (1.527-2.374) * | 0.000/75.9 | 0.491 (0.394-0.612) * | 0.000/77.4 | 0.314 (0.235-0.420) | 0.448/0.3 | 0.421 (0.278-0.637) * | 0.000/95.5 |
| NO | 19 (13223/15728) | 0.330 (0.156-0.698) * | 0.000/86.5 | 1.652 (1.237-2.207) * | 0.000/91.3 | 0.571 (0.418-0.779) * | 0.000/93.0 | 0.372 (0.191-0.725) * | 0.000/82.5 | 0.708 (0.512-0.981) * | 0.000/95.1 |
| **HWE** | | | | | | | | | | | |
| Compliant | 35 (17562/20898) | 0.303 (0.191-0.481) * | 0.000/78.6 | 1.785 (1.487-2.143) * | 0.000/87.4 | 0.526 (0.433-0.639) * | 0.000/77.4 | 0.349 (0.232-0.526) * | 0.000/72.5 | 0.541 (0.416-0.704) * | 0.000/95.9 |
| In Violation/NR | 4 (782/813) | 0.240 (0.108-0.532) | 0.365/0.0 | 1.359 (0.335-5.512) * | 0.000/87.6 | 0.661 (0.195-2.237) * | 0.000/85.6 | 0.260 (0.118-0.575) | 0.249/24.7 | 0.612 (0.273-1.374) * | 0.000/84.3 |

*(Continued)*

**Table 3.** (Continued)

| Variable | n (Cases/Controls) | CC vs. TT OR (95% CI) | Ph/I2(%) | CT vs. CC OR (95% CI) | Ph/I2(%) | CC vs. CT+TT OR (95% CI) | Ph/I2(%) | CT+CC vs. TT OR (95% CI) | Ph/I2(%) | C vs. T OR (95% CI) | Ph/I2(%) |
|---|---|---|---|---|---|---|---|---|---|---|---|
| **Sensitivity analysis** | | | | | | | | | | | |
| **Overall** | 27 (6696/7887) | 0.294 (0.230-0.375) | 0.545/0.0 | 1.727 (1.487-2.007) * | 0.000/65.3 | 0.540 (0.464-0.629) * | 0.000/68.5 | 0.334 (0.262-0.426) | 0.679/0.0 | 0.546 (0.393-0.759) * | 0.000/95.1 |
| **Ethnicity** | | | | | | | | | | | |
| Caucasian | 22 (6042/7098) | 0.300 (0.231-0.389) | 0.371/6.8 | 1.743 (1.503-2.022) * | 0.000/62.9 | 0.541 (0.464-0.631) * | 0.000/67.4 | 0.346 (0.267-0.448) | 0.494/0.0 | 0.562 (0.388-0.816) * | 0.000/95.9 |
| Asian | 3 (387/342) | 0.196 (0.042-0.907) | 0.898/0.0 | 1.956 (1.174-3.257) | 0.295/18.0 | 0.450 (0.275-0.736) | 0.360/2.1 | 0.210 (0.045-0.969) | 0.915/0.0 | 0.420 (0.264-0.666) | 0.431/0.0 |
| **Geographic region** | | | | | | | | | | | |
| Europe | 17 (4342/4725) | 0.324 (0.246-0.426) | 0.218/20.2 | 1.569 (1.303-1.891) * | 0.000/70.8 | 0.598 (0.495-0.723) | 0.000/73.7 | 0.368 (0.280-0.483) | 0.332/10.5 | 0.563 (0.354-0.897) | 0.000/96.8 |
| Asia | 5 (643/713) | 0.195 (0.049-0.779) | 0.975/0.0 | 2.249 (1.480-3.657) | 0.321/14.6 | 0.386 (0.248-0.599) * | 0.381/4.5 | 0.208 (0.052-0.829) | 0.981/0.0 | 0.696 (0.272-1.780) * | 0.005/73.5 |
| **Source of control** | | | | | | | | | | | |
| PB | 15 (3443/4450) | 0.358 (0.255-0.501) | 0.238/19.3 | 1.452 (1.165-1.811) * | 0.000/69.7 | 0.647 (0.517-0.809) * | 0.000/72.5 | 0.386 (0.276-0.541) | 0.335/10.6 | 0.803 (0.604-1.069) * | 0.000/86.8 |
| HB | 12 (3253/3437) | 0.250 (0.174-0.359) | 0.961/0.0 | 2.035 (1.804-2.295) | 0.349/9.9 | 0.459 (0.409-0.516) | 0.353/9.5 | 0.302 (0.211-0.433) | 0.936/0.0 | 0.335 (0.191-0.588) * | 0.000/96.3 |
| **Type of control** | | | | | | | | | | | |
| Healthy controls | 17 (3782/4234) | 0.307 (0.225-0.419) | 0.277/15.4 | 1.671 (1.329-2.100) * | 0.000/71.6 | 0.557 (0.442-0.702) * | 0.000/74.1 | 0.349 (0.256-0.475) | 0.388/5.7 | 0.460 (0.281-0.754) * | 0.000/95.9 |
| Non-diabetic controls | 10 (339/4333) | 0.275 (0.185-0.409) | 0.766/0.0 | 1.728 (1.540-1.938) | 0.031/51.1 | 0.523 (0.433-0.632) | 0.000/56.5 | 0.334 (0.224-0.499) | 0.814/0.0 | 0.704 (0.488-1.014) | 0.000/90.8 |
| **Egger's test** | | | | | | | | | | | |
| $P_E$ | | 0.402 | | 0.994 | | 0.908 | | 0.352 | | 0.875 | |

HB = hospital-based studies, PB = population-based studies, *=Random effects model.

**Table 4. Meta-analysis of the association of PTPN22 rs1310182 polymorphism with risk of T1D.**

| Variable | n (Cases/ Controls) | CC vs. TT | | CT vs.CC | | TT vs. CT+CC | | TT +CT vs. CC | | C vs. T | |
|---|---|---|---|---|---|---|---|---|---|---|---|
| | | OR (95% CI) | $P_h/I^2$(%) | OR (95% CI) | $P_h/I^2$(%) | OR (95% CI) | $P_h/I^2$(%) | OR (95% CI) | $P_h/I^2$(%) | OR (95% CI) | $P_h/I^2$(%) |
| **Overall** | 6 (842/801) | 0.560 (0.270-1.162) * | 0.000/80.3 | 1.325 (0.740-2.371) * | 0.004/80.5 | 0.726 (0.290-1.819) * | 0.000/93.5 | 1.465 (0.813-2.640) * | 0.000/83.2 | 0.755 (0.510-1.119) * | 0.000/85.5 |
| **Ethnicity** | | | | | | | | | | | |
| Caucasian | 4 (519/451) | 0.769 (0.331-1.785) * | 0.005/76.5 | 1.306 (0.473-3.601) * | 0.000/87.7 | 1.058 (0.436-2.570) * | 0.000/88.3 | 1.294 (0.459-3.382) * | 0.000/87.7 | 0.918 (0.593-1.422) * | 0.001/80.7 |
| **Source of control** | | | | | | | | | | | |
| HB | 4 (552/552) | 0.465 (0.177-1.221) * | 0.000/83.6 | 1.562 (0.635-3.844) * | 0.000/86.0 | 0.950 (0.447-2.016) * | 0.000/86.6 | 1.759 (0.713-4.342) * | 0.000/87.6 | 0.704 (0.402-1.233) * | 0.000/89.4 |
| **Blinding and/or Quality control** | | | | | | | | | | | |
| YES | 4 (559/522) | 0.465 (0.177-1.221) * | 0.000/83.6 | 1.562 (0.635-3.844) * | 0.000/86.0 | 1.759 (0.713-4.342) * | 0.000/87.6 | 0.950 (0.447-2.016) * | 0.000/86.6 | 0.704 (0.402-1.233) * | 0.000/89.4 |
| **Sensitivity analysis** | | | | | | | | | | | |
| **Overall** | 4 (559/522) | 0.465 (0.177-1.221) * | 0.000/83.6 | 1.562 (0.635-3.844) * | 0.000/86.0 | 1.759 (0.713-4.342) * | 0.000/87.6 | 0.950 (0.447-2.016) * | 0.000/86.6 | 0.704 (0.402-1.233) * | 0.000/89.4 |
| **Ethnicity** | | | | | | | | | | | |
| Caucasian | 3 (420/351) | 0.619 (0.207-1.854) * | 0.006/80.4 | 1.687 (0.433-6.569) * | 0.000/90.6 | 1.636 (0.445-6.015) * | 0.000/90.7 | 0.871 (0.291-2.607) * | 0.000/90.2 | 0.845 (0.471-1.513) * | 0.001/85.8 |
| **Source of control** | | | | | | | | | | | |
| HB | 4 (559/522) | 0.465 (0.177-1.221) * | 0.000/83.6 | 1.562 (0.635-3.844) * | 0.000/86.0 | 1.759 (0.713-4.342) * | 0.000/87.6 | 0.950 (0.447-2.016) * | 0.000/86.6 | 0.704 (0.402-1.233) * | 0.000/89.4 |
| **Egger' s test** | | | | | | | | | | | |
| $P_E$ | | 0.854 | | 0.616 | | 0.726 | | 0.720 | | 0.683 | |

HB=hospital-based studies, PB=population-based studies, *=Random effects model.

To assess the stability of the included studies, sensitivity analyses were performed using a case-by-case exclusion method. For *PTPN22* rs2476601, excluding the study by Chelala et al [31] reduced heterogeneity from $I^2$ = 77.5% to 0.0% (P = 0.594). In contrast, excluding individual studies for *PTPN22* rs1310182 did not significantly affect the results. To exclude low-quality studies with HWD [44,45,54,56] and quality scores ≤ 12 [14,19,26,27,31,38–40,44,45,49,50], we conducted additional subgroup analyses based on ethnicity and study quality to further explain the heterogeneity (Fig 2). In Caucasian and Asian populations, rs2476601 was negatively associated with T1D risk (CC vs. TT: Caucasian OR = 0.254, 95% CI = 0.211–0.305 Asian OR = 0.277, 95% CI = 0.123–0.622). However, in mixed-race populations, rs2476601 showed no association with T1D risk (CC vs. TT: OR = 0.473, 95% CI = 0.055–4.094). Subsequently, when only high-quality studies and studies in HWE were included, the OR for the overall study and subgroup analyses remained largely unaffected.

## Publication bias

In this study, publication bias was assessed using the Begg funnel plot and Egger's test. When bias was detected, the results were adjusted using the non-parametric trim-and-fill method. The funnel plots for all models (Fig 3) appeared roughly symmetric, and Egger's test results were all above 0.05 (rs2476601: CC vs. TT: P = 0.402; CT vs. CC: P = 0.994; CC vs. CT+TT: P = 0.908; CT+CC vs. TT: P = 0.352 and C vs. T: P = 0.875, Table 3; rs1310182: CC vs. TT: P = 0.854; CT vs. CC: P = 0.616; CC vs. CT+TT: P = 0.726; CT+CC vs. TT: P = 0.720 and C vs. T: P = 0.683, Table 4), indicating no evidence of publication bias.

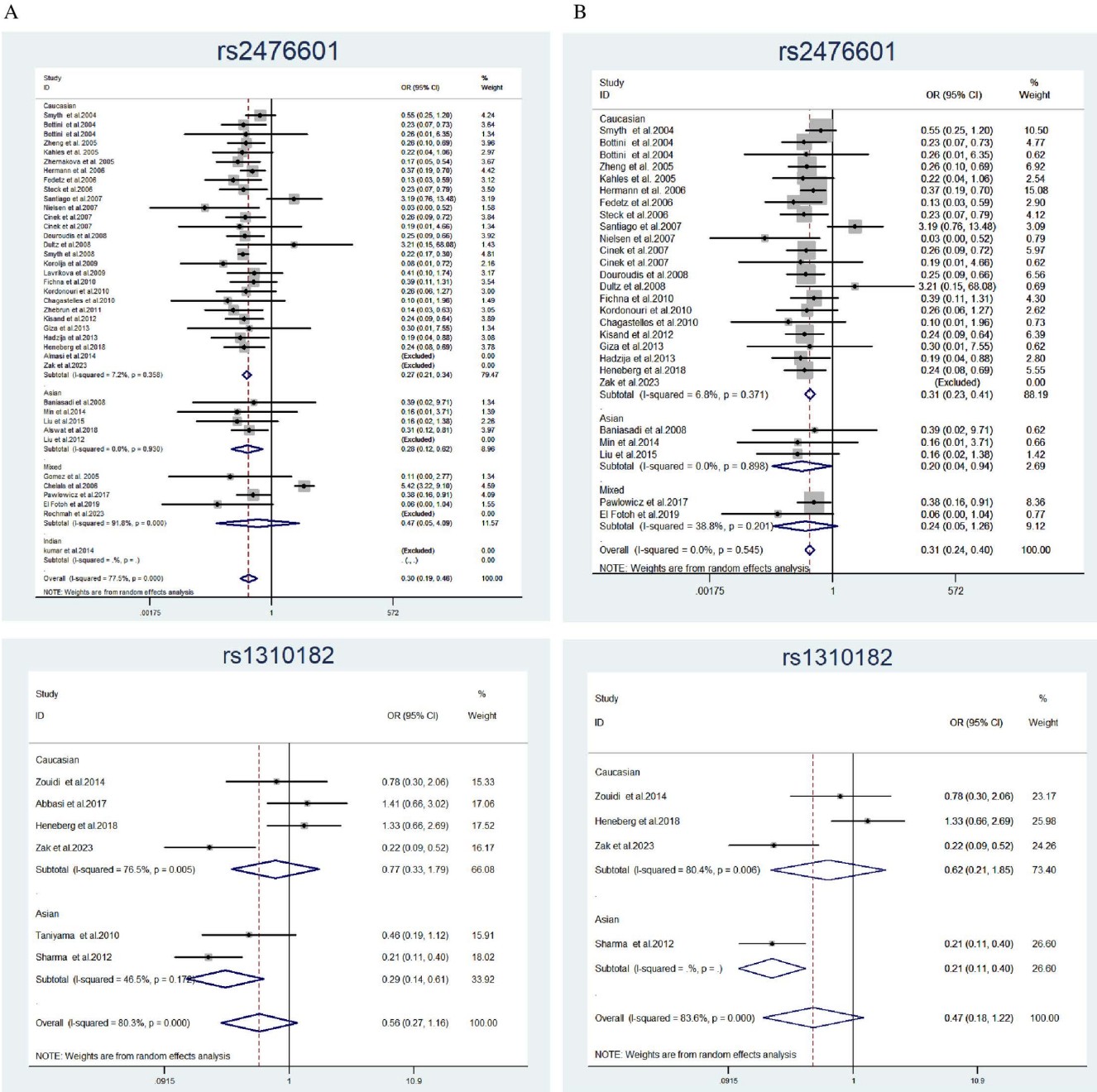

**Fig 2. Forest plot of subgroup analysis on PTPN22 rs2476601 and rs1310182 polymorphisms (CC vs.TT) by ethnicity and study quality.** [A: **Ethnicity (Caucasian, Asian, Mixed and Indian)**; B: **Study quality (HWE and Quality score>12)**].

## Credibility of the identified genetic associations

To assess the credibility of the study, we applied FPRP, BEDP, and Venice criteria to support a high-confidence association. The criteria were as follows: (1) statistically significant associations observed in at least two genetic models, with. P-values for the Z-test < 0.05; (2) FPRP < 0.2 and BFDP < 0.8; (3) statistical power > 0.8; and (4) $I^2$ < 50%. A lower standard of "less

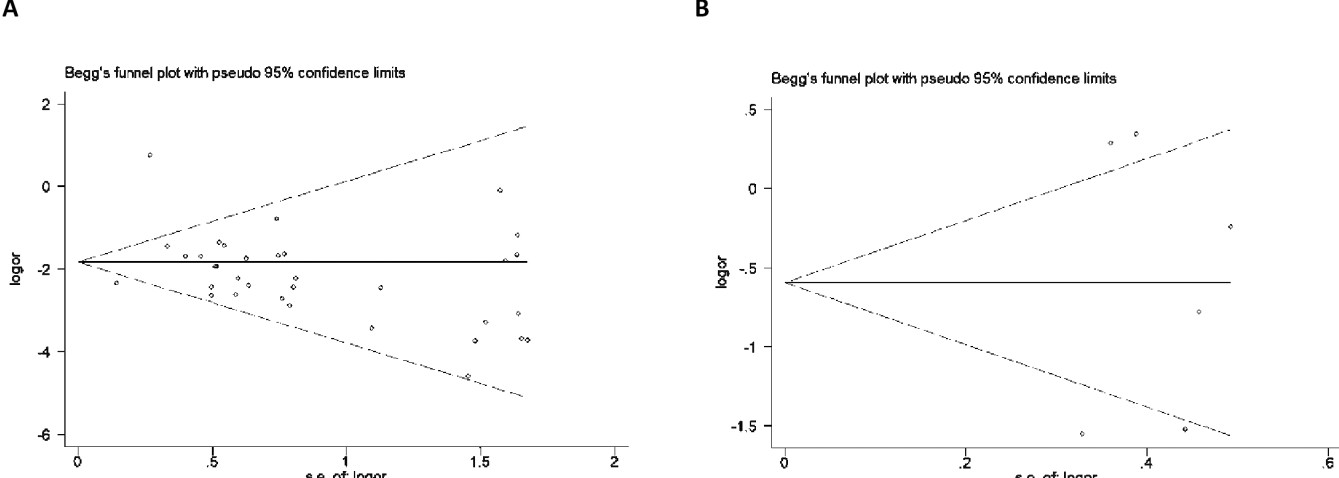

**Fig 3. Begg's funnel plot of PTPN22 polymorphisms and T1D susceptibility in the general population (CTvs.TT).**

credible certainty" is applied when the following conditions are met: (1) at least one genetic model had a P-value ≥ 0.05.; (2) statistical power was between 50% and 79%, or FPRP > 0.2 or $I^2$ > 50%. Table 5 presents the confidence assessment results for the *PTPN22* rs2476601 polymorphism and its association with T1D susceptibility. The meta-analysis indicated that, in Caucasians, the rs2476601 polymorphism was associated with an increased T1D risk, yielding a "positive result with a high degree of confidence". In contrast, a "less credible positive result" was found in Asians. "Highly credible positive results" were found in hospital-based studies, whereas no positive associations were identified in population-based studies. Given the limited sample size, the observed effect for rs1310182 may not be significant. Consequently, these findings carry a high degree of uncertainty, and the calculation of FPRP and BFDP values was not feasible.

## Discussion

T1D is a clinical form of diabetes mellitus characterized by an absolute deficiency of insulin due to the destruction of pancreatic β-cells [1]. It is often associated with a predisposition to ketosis and results from a complex interaction of genetic and environmental factors [64]. Extensive evidence highlights the varying susceptibility to T1D across different populations, with individual genetic factors playing a critical role in the disease's pathogenesis [6]. In addition to the HLA region, the *PTPN22* gene has emerged as a key genetic factor in autoimmune susceptibility [64]. The *PTPN22* gene encodes a protein tyrosine phosphatase that is essential for T cell receptor signaling and the maintenance of immune tolerance, thus playing a pivotal role in immune regulation.[65,66]. Notably, variations in the *PTPN22* gene, particularly the R620W polymorphism, have been linked to an increased risk of autoimmune disorders, including T1D. This polymorphism may disrupt normal immune function, promoting autoimmunity in individuals with T1D. Recent studies emphasize the significant role of *PTPN22* in T1D susceptibility, yet inconsistencies remain in the literature regarding the association between *PTPN22* SNPs and T1D risk [13,14,34,35]. This study systematically reviews 41 articles to provide evidence supporting the association between the *PTPN22* rs247660 and rs1310182 polymorphisms and T1D susceptibility.

**Table 5. Credibility of the current meta-analysis.**

| Variables | Model | OR (95% CI) | I2 (%) | Statistical power | Credibility | |
|---|---|---|---|---|---|---|
| | | | | | Prior probability of 0.001 | |
| | | | | | FPRP | BFDP |
| *PTPN22* rs2476601 | | | | | | |
| **Overall** | CC vs.TT | 0.298 (0.193-0.459) | 77.5 | 0.000 | 0.029 | 0.001 |
| | CT vs.CC | 1.777 (1.484-2.128) | 87.1 | 0.033 | <0.001 | <0.001 |
| | CC vs. CT+TT | 0.528 (0.437-0.639) | 89.3 | 0.008 | 0.025 | <0.001 |
| | CT+CC vs. TT | 0.341 (0.231-0.502) | 71.3 | 0.001 | 0.611 | 0.005 |
| | C vs. T | 0.547 (0.426-0.701) | 95.5 | 0.059 | 0.031 | 0.084 |
| **Caucasian** | CC vs.TT | 0.254 (0.211-0.305) | 7.2 | 0.422 | 0.001 | <0.001 |
| | CT vs.CC | 1.764 (1.553-2.004) | 68.6 | 0.954 | 0.006 | <0.001 |
| | CC vs. CT+TT | 0.534 (0.470-0.607) | 70.7 | 0.952 | 0.007 | <0.001 |
| | CT+CC vs. TT | 0.295 (0.246-0.355) | 0.0 | 0.402 | 0.014 | 0.994 |
| | C vs. T | 0.553 (0.429-0.714) | 0.0 | 0.076 | 0.068 | 0.197 |
| **Asian** | CC vs.TT | 0.277 (0.123-0.622) | 0.0 | 0.017 | 0.917 | 0.879 |
| | CT vs.CC | 2.921 (2.027-4.207) | 43.4 | <0.001 | 0.047 | 0.001 |
| | CC vs. CT+TT | 0.356 (0.229-0.554) | 22.5 | 0.003 | 0.146 | 0.031 |
| | CT+CC vs. TT | 0.306 (0.137-0.683) | 0.0 | 0.029 | 0.993 | 0.991 |
| | C vs. T | 0.333 (0.245-0.454) | 0.0 | <0.001 | 0.001 | <0.001 |
| **PB** | CC vs.TT | 0.345 (0.102-0.656) | 84.7 | 0.022 | 0.839 | 0.795 |
| | CT vs.CC | 1.533 (1.190-1.974) | 91.0 | 0.433 | 0.681 | 0.961 |
| | CC vs. CT+TT | 0.614 (0.469-0.804) | 92.6 | 0.275 | 0.124 | 0.525 |
| | CT+CC vs. TT | 0.382 (0.215-0.679) | 80.5 | 0.029 | 0.973 | 0.969 |
| | C vs. T | 0.724 (0.549-0.955) | 94.6 | 0.720 | 0.969 | 0.997 |
| **HB** | CC vs.TT | 0.257 (0.183-0.361) | 0.0 | <0.001 | <0.001 | <0.001 |
| | CT vs.CC | 2.121 (1.888-2.382) | 43.3 | 0.989 | 0.006 | <0.001 |
| | CC vs. CT+TT | 0.427 (0.363-0.502) | 40.5 | 0.006 | 0.478 | <0.001 |
| | CT+CC vs. TT | 0.309 (0.221-0.433) | 0.0 | <0.001 | 0.002 | <0.001 |
| | C vs. T | 0.349 (0.213-0.572) | 95.5 | 0.005 | 0.853 | <0.001 |
| HWE and Quality score > 12 | | | | | | |
| **Overall** | CC vs.TT | 0.294 (0.230-0.375) | 0.0 | 0.079 | 0.067 | <0.001 |
| | CT vs.CC | 1.727 (1.487-2.007) | 65.3 | 0.033 | <0.001 | <0.001 |
| | CC vs. CT+TT | 0.540 (0.464-0.629) | 68.5 | 0.003 | <0.001 | <0.001 |
| | CT+CC vs. TT | 0.334 (0.262-0.426) | 0.0 | 0.101 | 0.052 | <0.001 |
| | C vs. T | 0.546 (0.393-0.759) | 95.1 | 0.117 | 0.730 | 0.901 |
| **Caucasian** | CC vs.TT | 0.300 (0.231-0.389) | 6.8 | 0.056 | 0.091 | <0.001 |
| | CT vs.CC | 1.743 (1.503-2.022) | 62.9 | 0.024 | <0.001 | <0.001 |
| | CC vs. CT+TT | 0.541 (0.464-0.631) | 67.4 | 0.004 | <0.001 | <0.001 |
| | CT+CC vs. TT | 0.346 (0.267-0.448) | 0.0 | 0.071 | 0.073 | <0.001 |
| | C vs. T | 0.562 (0.388-0.816) | 95.9 | 0.185 | 0.930 | 0.981 |
| **Asian** | CC vs.TT | 0.196 (0.042-0.907) | 0.0 | 0.089 | 0.984 | 0.983 |
| | CT vs.CC | 1.956 (1.174-3.257) | 18.0 | 0.154 | 0.985 | 0.994 |
| | CC vs. CT+TT | 0.450 (0.275-0.736) | 2.1 | 0.059 | 0.712 | 0.782 |
| | CT+CC vs. TT | 0.210 (0.045-0.969) | 0.0 | 0.069 | 0.998 | 0.998 |
| | C vs. T | 0.420 (0.264-0.666) | 0.0 | 0.025 | 0.901 | 0.888 |
| **PB** | CC vs.TT | 0.358 (0.255-0.501) | 19.3 | 0.000 | 0.001 | <0.001 |
| | CT vs.CC | 1.452 (1.165-1.811) | 69.7 | 0.614 | 0.604 | 0.963 |
| | CC vs. CT+TT | 0.647 (0.517-0.809) | 72.5 | 0.396 | 0.032 | 0.311 |

*(Continued)*

**Table 5.** (Continued)

| Variables | Model | OR (95% CI) | I2 (%) | Statistical power | Credibility | |
|---|---|---|---|---|---|---|
| | | | | | Prior probability of 0.001 | |
| | | | | | FPRP | BFDP |
| | CT+CC vs. TT | 0.386 (0.276-0.541) | 10.6 | 0.001 | 0.041 | 0.003 |
| **HB** | CC vs.TT | 0.250 (0.174-0.359) | 0.0 | <0.001 | <0.001 | <0.001 |
| | CT vs.CC | 2.035 (1.804-2.295) | 9.9 | 0.981 | 0.006 | <0.001 |
| | CC vs. CT+TT | 0.459 (0.409-0.516) | 9.5 | 0.019 | 0.230 | <0.001 |
| | CT+CC vs. TT | 0.302 (0.211-0.433) | 0.0 | <0.001 | 0.009 | <0.001 |
| | C vs. T | 0.335 (0.191-0.588) | 96.3 | 0.007 | 0.889 | 0.875 |

HB = hospital-based studies, PB = population-based studies, HWE= Hardy-Weinberg equilibrium.

In conclusion, the rs2476601 polymorphism was found to significantly increase the risk of T1D, especially in Caucasian and Asian populations. Subgroup analyses, including studies with matched controls, blinded quality controls, and tests for HWE, consistently demonstrated a significant association between the rs2476601 polymorphism and T1D risk. Given the substantial heterogeneity observed in the overall analysis, we performed additional subgroup analyses by ethnicity and sensitivity analyses, The rs2476601 polymorphism was found to be significantly associated with T1D risk in both the overall analysis and in Caucasians and Asians. Regional differences must be considered when interpreting genetic associations within Caucasian populations. Consequently, we excluded mixed-race populations and performed subgroup analyses based on geographic region (Eastern, Western, Northern, Southern, and Central Europe). Forest plots and heterogeneity analyses revealed significant regional variations in effect sizes. Heterogeneity was highest in Southern Europe ($I^2$ = 63.2%, P = 0.028) and moderate in Western Europe ($I^2$ = 59.3%, P = 0.086), while Eastern, Northern, and Central Europe exhibited minimal heterogeneity ($I^2$ = 0.0–10.5%, P > 0.3). The overall analysis demonstrated a significant correlation (z = 8.60, P < 0.001), consistent with findings in most regions. However, while the overall analysis showed a significant correlation, the association in Southern Europe was not statistically significant (z = 1.23, P = 0.217), underscoring the limitations of generalizing these results to the broader Caucasian population. These regional disparities may arise from factors such as genetic drift, founder effects, or distinct environmental influences. This emphasizes the importance of accounting for geographic variation in genetic studies and cautions against overgeneralization, while also reinforcing the robustness of the associations observed in most regions. Previous meta-analyses have suggested that the *PTPN22* rs2476601 polymorphism is a risk factor for T1D in Caucasians [7–10]. Our findings are consistent with these studies and reinforce the conclusion that this polymorphism contributes to an increased T1D susceptibility in Caucasians, supporting the reliability of our results. Regarding the rs1310182 polymorphism, pooled data from several independent studies revealed no significant association with T1D risk.

Given the large volume of genomic data being generated, we applied a comprehensive correction using the FPRP, BFDP, and Venice criteria. Our findings showed no correlation between the *PTPN22* rs2476601 polymorphism and T1D risk across four genetic models, with $I^2$ > 75%. However, we observed correlations between the *PTPN22* rs2476601 polymorphism and T1D risk in the Caucasian and Asian subgroups based on ethnicity, as well as in hospital-based studies. The correlation with T1D risk was weaker in population-based studies. After correcting the sensitivity analysis results with the FPRP, BFDP, and Venice criteria,

significant correlations emerged in the overall analysis, though the significance for Asians markedly declined. This suggests that integrating quantitatively synthesized and low-quality data may affect the reliability of the results. The findings related to Caucasians and HB sources are considered a "high-confidence positive result" for T1D susceptibility. In contrast, the results associated with Asians are regarded as "false positives" with low confidence, while other results are deemed unreliable. Further high-quality studies are needed to draw definitive conclusions.

A comparison of the characteristics of this study with those of previous meta-analyses is provided in S3 Table. This meta-analysis includes 45 studies, significantly expanding upon earlier work. Of these, 39 studies examined the *PTPN22* rs2476601 polymorphism, and 6 focused on the rs1310182 polymorphism. This study utilized a larger sample size than previous analysis and incorporated the most recent studies from 2023, whereas earlier studies were predominantly published before 2013. A review of past studies shows that Bottini first investigated the *PTPN22* gene polymorphism and its association with T1D in 2002 [67], followed by a 2004 study showing an increased T1D risk in North America and Sardinia [6]. Subsequently, a 2006 meta-analysis by Lee linked the *PTPN22* SNP to autoimmune diseases such as rheumatoid arthritis, systemic lupus erythematosus, Graves' disease, T1D, and juvenile idiopathic arthritis [68]. Later analyses suggested that the *PTPN22* rs2476601 polymorphism may be a T1D risk factor in Caucasians, which aligns with our findings [7–10]. However, previous meta-analyses had several significant flaws. For example, only two studies stratified analyses by ethnicity, revealing inconsistencies across studies from different countries [7,10]. Additionally, data overlap, the inclusion of unnecessary data, and the omission of study quality assessments likely biased the results by incorporating low-quality studies [10]. Furthermore, one analyse ignored HWE, risking selection bias or genotyping errors [10]. All previous studies lacked sensitivity assessments and employed incomplete search strategies. Different genetic models were used, with some analyses comparing three models [7,9,10] and others four [8], which may have caused false negatives. Importantly, previous meta-analyses failed to assess the probability of false-positive results, raising concerns about their credibility. Notably, Wakefield (2007) proposed a more accurate Bayesian approach for detecting genetic errors in epidemiologic surveys. Many factors can contribute to error and bias, such as genotyping errors and phenotypic misclassification, with statistical power being a key factor. A large body of evidence (statistical power >80%) can achieve high statistical significance or reduce false discovery rates [69]. By neglecting confidence in the results, previous studies may have led to unreliable conclusions.

To better understand the relationship between the *PTPN22* polymorphism and the risk of developing T1D, we conducted an updated meta-analysis. In this analysis, we incorporated a larger sample size and included quality assessments, as well as HWE tests for the relevant studies. Additionally, meta-regression analyses were performed, considering nine factors: ethnicity, sample size, control source, case source, HWE, family history, association assessment with appropriate statistics, adjustment for confounders, and quality scores. The goal was to identify potential sources of heterogeneity. Stratified analyses were also conducted based on the studies' epidemiological characteristics, with particular focus on sensitivity analyses. These analyses prioritized high-quality studies with elevated composite scores and HWE conformity to minimize random errors and confounding biases. Finally, significant findings were adjusted using the FPRP test, the BFDP test, and the Venice criteria.

Despite employing several strategies to address the limitations of previous studies, this study has some inherent limitations. First, only published articles were included, which may have led to the omission of relevant data. Second, discrepancies were observed between the SNP frequencies in our study and those reported in databases such as HapMap and the 1000

Genomes Project (S4 Table). While these databases provide valuable population-level allele frequencies, they reflect broad ethnic classifications and general populations rather than cohorts specifically stratified for T1D. Genetic drift, founder effects, and subpopulation differences within ethnic groups may explain variations in allele frequencies. For instance, the higher T allele frequency of rs2476601 in our case group aligns with its role as a T1D susceptibility locus, while its slight enrichment in controls could reflect undiagnosed autoimmune conditions or environmental selection pressures. Differences in sample size, genotyping methodologies, and incomplete exclusion of subclinical cases in control groups may further contribute to heterogeneity, affecting the accuracy and interpretability of results. Third, the limited number of studies with partial subgroup analyses and the lack of relevant data from African and Indian populations hindered a comprehensive evaluation of the impact of these polymorphisms on the pathogenesis of T1D in these populations, highlighting the need for broader, multi-ethnic investigations. Fourth, the selection of controls varied across studies – where some explicitly excluded T1D cases, while others used asymptomatic individuals as controls. This variation may have introduced misclassification bias due to incomplete exclusion of potential diabetes or autoimmune cases. Fifth, although we accounted for geographical heterogeneity in allele frequencies among Caucasian populations, the lack of detailed environmental and lifestyle data limits our ability to fully explore gene-environment interactions, which are likely to play a significant role in T1D susceptibility. Finally, our analyses did not account for potential confounders, including age, sex, seasonal variations, dietary factors, and study design. Therefore, future studies should prioritize larger, more diverse cohorts to validate these findings and elucidate the genetic and environmental interplay in T1D susceptibility. Additionally, harmonizing control group criteria, improving subpopulation stratification, and integrating data on confounders will enhance the precision of meta-analyses in this field.

## Conclusion

A comprehensive analysis of 45 studies concluded that the *PTPN22* rs2476601 polymorphism is associated with an increased susceptibility to T1D in both Caucasian and Asian populations, indicating a positive correlation with T1D. Subgroup and sensitivity analyses further support the robustness of these findings, although the persistence of significant heterogeneity. Meta-regression analysis identifies ethnicity and genotyping quality as major sources of variability. However, the association between this polymorphism and T1D in Asian populations should be interpreted with caution due to potential for false positives. In contrast, the rs1310182 polymorphism does not show a significant association with T1D. Further studies involving diverse populations are required to confirm these associations. Additionally, investigating the biological relevance of functional SNPs in regulating T1D activity and gene expression is crucial, as it will enhance our understanding of their role in the disease's pathogenesis.

## Supporting information

**S1 Table. Scale for quality assessment of molecular association studies of T1D.**
(DOCX)

**S2 Table. Quality assessment of included studies for assessing the quality of case control studies.**
(DOCX)

**S3 Table. Included studies of PTPN22 polymorphisms in T1D within the meta-analyses.**
(DOCX)

**S4 Table. Risk Allele Frequency Comparisons (Current Study, HapMap, 1000 Genomes).**
(DOCX)

**S5 Text. PRISMA 2020 checklist.**
(DOC)

**S6 Text. plosone-checklist.**
(PDF)

**S1 File. Raw data and literature search.**
(XLS)

## Author contributions

**Conceptualization:** Yu - Su.

**Data curation:** Yu - Su, Xue - Li, Pei-dong Wu, Peng-fei Fang.

**Formal analysis:** Xiao-feng He.

**Methodology:** Xiao-feng He.

**Supervision:** Fei-fei Wu.

**Validation:** Yu - Su, Yu-Long Zhang.

**Writing – original draft:** Yu - Su, Xue - Li, Pei-dong Wu.

**Writing – review & editing:** Yu - Su, Xue - Li, Pei-dong Wu.

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
