## [Decision Letter · Decision Letter 0]

26 Dec 2024

PONE-D-24-50318The Association between PTPN22 SNPs and Susceptibility to Type 1 Diabetes: An updated meta-analysisPLOS ONE

Dear Dr. Wu,

Thank you for submitting your manuscript to PLOS ONE. After careful consideration, we feel that it has merit but does not fully meet PLOS ONE’s publication criteria as it currently stands. Therefore, we invite you to submit a revised version of the manuscript that addresses the points raised during the review process.

We look forward to receiving your revised manuscript.

Kind regards,

Petr Heneberg

Academic Editor

PLOS ONE

2. We note that your Data Availability Statement is currently as follows: [We gratefully acknowledge the contributors to all the original studies included in this meta-analysis.]

4. Please include a new copy of Tables in your manuscript; the current table is difficult to read. Please follow the link for more information: https://blogs.plos.org/plos/2019/06/looking-good-tips-for-creating-your-plos-figures-graphics/

5. As required by our policy on Data Availability, please ensure your manuscript or supplementary information includes the following:

Reviewers' comments:

Reviewer's Responses to Questions

**Comments to the Author**

1. Is the manuscript technically sound, and do the data support the conclusions?

Reviewer #1: Partly

Reviewer #2: Yes

2. Has the statistical analysis been performed appropriately and rigorously? 

Reviewer #1: Yes

Reviewer #2: Yes

3. Have the authors made all data underlying the findings in their manuscript fully available?

Reviewer #1: Yes

Reviewer #2: Yes

4. Is the manuscript presented in an intelligible fashion and written in standard English?

Reviewer #1: No

Reviewer #2: Yes

5. Review Comments to the Author

Reviewer #1: This manuscript reports and interesting meta-analysis of the association of PTPN22 SNPs and susceptibility to type 1 diabetes.

The manuscript should be completed by an indept analysis when possible of the frequency of the SNPs in different populations where these data have not been evaluated

The manuscript requires revision by a native english speaker. I would revise the sentence in the abstract 'the findings regarding the relationship between SNPs and T1D susceptibility have been inconsistent'. Studies indeed were 'not conclusive in different populations' while in some such as the Italian or caucasian populations the frequency has been extimated and found significant in diabetics at least for the C1858T SNP

Reviewer #2: The Manuscript sounds interesting and cover all aspects and I would suggest a minor revision to improve the discussion section more with slightly grammarly and English improvment while the data is correctly addressed and conclsuion you can add 2 3 lines more.

6. PLOS authors have the option to publish the peer review history of their article (what does this mean? ). If published, this will include your full peer review and any attached files.

**Do you want your identity to be public for this peer review?** For information about this choice, including consent withdrawal, please see our Privacy Policy .

Reviewer #1: No

Reviewer #2: **Yes: ** Sammra Maqsood

---

## [Author Response · Author response to Decision Letter 0]

19 Feb 2025

Dear PLOS ONE Staff,

On behalf of all co-authors, I sincerely appreciate your constructive feedback and the opportunity to revise our manuscript titled "The Association between PTPN22 SNPs and Susceptibility to Type 1 Diabetes: An updated meta-analysis" (manuscript number: PONE-D-24-50318). We have carefully addressed each point raised in your review and detailed our responses below. Revised sections are marked in the manuscript, and supplementary files have been updated accordingly.

Response: Thank you for providing the Author Formatting Checklist to ensure that our paper meets PLOS ONE's typesetting requirements for References, Tables, and Figures. We will carefully review the checklist and make the necessary adjustments to our submission files accordingly.

2.We note that your Data Availability Statement is currently as follows: [We gratefully acknowledge the contributors to all the original studies included in this meta-analysis.]

Response: We confirm that our submission includes all raw data required to replicate the results of our study. The data has been provided in the supplementary materials as per the journal’s requirements(Supporting Information files).

3.PLOS requires an ORCID iD for the corresponding author in Editorial Manager on papers submitted after December 6th, 2016. Please ensure that you have an ORCID iD and that it is validated in Editorial Manager. To do this, go to ‘Update my Information’ (in the upper left-hand corner of the main menu), and click on the Fetch/Validate link next to the ORCID field. This will take you to the ORCID site and allow you to create a new iD or authenticate a pre-existing iD in Editorial Manager.

Response: The ORCID iD of the corresponding author (Fei-fei Wu: 0009-0000-0333-9833) has been added to the title page and verified in Editorial Manager.

4.Please include a new copy of Tables in your manuscript; the current table is difficult to read. Please follow the link for more information: https://blogs.plos.org/plos/2019/06/looking-good-tips-for-creating-your-plos-figures-graphics/

Response: All tables (Tables 1-5) have been reformatted using Arial font for clarity. Headings have been bolded for better differentiation and improved readability. The updated tables are included in the manuscript and in the supplementary document.

5.As required by our policy on Data Availability, please ensure your manuscript or supplementary information includes the following:

Response: Thank you for reviewing our work and providing valuable feedback. We fully acknowledge the importance of including a list of excluded studies to enhance transparency and reproducibility. Excel files containing all relevant data, such as the names of data extractors, extraction dates, eligibility criteria, and the necessary information for replication, are provided in the Supporting Information files. Furthermore, a table summarizing the risk of bias and quality assessments for each study or outcome is included (Table S2). The manuscript also contains a detailed explanation of the methods used to address missing data.

6.The manuscript should be completed by an indept analysis when possible of the frequency of the SNPs in different populations where these data have not been evaluated

Response: Thank you for your valuable feedback on our study. We noted the observed discrepancies between the SNP frequencies in our study and those reported in the HapMap and 1000 Genomes Project databases (Supplemental Table 4). We believe these discrepancies may stem from several factors: First, HapMap primarily reflects population frequencies under broad ethnic classifications, whereas our current study focuses on specific case-control cohorts. Genetic drift, founder effects, and subpopulation differences within ethnic groups may contribute to variations in allele frequencies. Second, the higher T allele frequency observed in our case group supports the role of rs2476601 as a susceptibility locus for type 1 diabetes (T1D), while the slight enrichment in the control group may suggest the presence of undiagnosed autoimmune conditions. Additionally, differences in sample size, genotyping methodologies, and environmental selection pressures could further explain the observed frequency variations. We agree with your suggestion that future studies involving larger and more diverse populations are needed to validate our findings and to further elucidate the genetic basis of T1D susceptibility. We have incorporated these explanations into the Discussion section and addressed additional limitations to enhance the clarity and context of our findings. We sincerely appreciate your constructive comments and thank you again for your insights.

7.The manuscript requires revision by a native english speaker. I would revise the sentence in the abstract 'the findings regarding the relationship between SNPs and T1D susceptibility have been inconsistent'. Studies indeed were 'not conclusive in different populations' while in some such as the Italian or caucasian populations the frequency has been extimated and found significant in diabetics at least for the C1858T SNP.

Response: The manuscript has been thoroughly edited by a native English speaker. The abstract sentence now reads: "Studies investigating the relationship between PTPN22 rs2476601 and T1D risk have consistently demonstrated an association in certain populations, whereas research on rs1310182 has yielded conflicting and less conclusive results. "

Additional grammatical improvements have been made throughout the manuscript, and track changes have been used to mark edits for clarity.

8.The Manuscript sounds interesting and cover all aspects and I would suggest a minor revision to improve the discussion section more with slightly grammarly and English improvment while the data is correctly addressed and conclsuion you can add 2 3 lines more.

Response: We appreciate the positive feedback. The Discussion section has been revised for clarity and grammatical accuracy. We have also expanded the Conclusion section to further emphasize the significance of our findings. Specifically, we added 2–3 lines to highlight the potential implications of our study for understanding genetic contributions to T1D and the importance of population-specific research.

9.PLOS authors have the option to publish the peer review history of their article . If published, this will include your full peer review and any attached files.

Response: We understand this option and will consider publishing the peer review history after the final acceptance of the manuscript.

10.While revising your submission, please upload your figure files to the Preflight Analysis and Conversion Engine (PACE) digital diagnostic tool, https://pacev2.apexcovantage.com/. PACE helps ensure that figures meet PLOS requirements. To use PACE, you must first register as a user. Registration is free. Then, login and navigate to the UPLOAD tab, where you will find detailed instructions on how to use the tool. If you encounter any issues or have any questions when using PACE, please email PLOS at figures@plos.org. Please note that Supporting Information files do not need this step.

Response: We thank you for suggesting that we use the PACE tool to ensure that our data meets the technical requirements of PLOS ONE. Following the instructions provided, we have performed the necessary evaluation and conversion of our graphical files to meet the journal's specifications.

Thank you for your detailed feedback and guidance. We hope that the revised manuscript meets the journal’s requirements. Please do not hesitate to contact us if further modifications are required.

Kind regards,

Fei-fei Wu

---

## [Decision Letter · Decision Letter 1]

28 Feb 2025

PONE-D-24-50318R1The Association between PTPN22 SNPs and Susceptibility to Type 1 Diabetes: An updated meta-analysisPLOS ONE

Dear Dr. Wu,

Thank you for submitting your manuscript to PLOS ONE. After careful consideration, we feel that it has merit but does not fully meet PLOS ONE’s publication criteria as it currently stands. Therefore, we invite you to submit a revised version of the manuscript that addresses the points raised during the review process.

We look forward to receiving your revised manuscript.

Kind regards,

Petr Heneberg

Academic Editor

PLOS ONE

Journal Requirements:

Additional Editor Comments:

- Tables 1 and 2 - Zak et al. 2023 examined patients from Armenia, not from America.

- Fig 2 - state clearly the reasons for exlusion of studies explicitly mentioned in the figure.

- Fig. 1 - typo "titlis".

- The whole study needs to better reflect the geographic gradient of the study polymorphisms within the Caucasian populations. There are enormous differences in the prevalence of the study rs in general and T1DM Caucasian populations both in the N-S and E-W directions throughout Europe. It is described the best for rs2476601. These differences are enormous and prevent generalization of conclusions for the whole Caucasian population.

- I cannot open the supplementary files. Please double-check their integrity and reupload them.

Reviewers' comments:

Reviewer's Responses to Questions

**Comments to the Author**

1. If the authors have adequately addressed your comments raised in a previous round of review and you feel that this manuscript is now acceptable for publication, you may indicate that here to bypass the “Comments to the Author” section, enter your conflict of interest statement in the “Confidential to Editor” section, and submit your "Accept" recommendation.

Reviewer #1: All comments have been addressed

Reviewer #2: All comments have been addressed

2. Is the manuscript technically sound, and do the data support the conclusions?

Reviewer #1: Yes

Reviewer #2: Yes

3. Has the statistical analysis been performed appropriately and rigorously? 

Reviewer #1: Yes

Reviewer #2: Yes

4. Have the authors made all data underlying the findings in their manuscript fully available?

Reviewer #1: Yes

Reviewer #2: Yes

5. Is the manuscript presented in an intelligible fashion and written in standard English?

Reviewer #1: Yes

Reviewer #2: Yes

6. Review Comments to the Author

Reviewer #1: the manuscript can be accepted in the present revised version. The authors have satisfactorely replied to the reviewers'comments and criticism

Reviewer #2: The author addressed all the comments and I highly recommend for the submission and all the points are addressed.

7. PLOS authors have the option to publish the peer review history of their article (what does this mean? ). If published, this will include your full peer review and any attached files.

**Do you want your identity to be public for this peer review?** For information about this choice, including consent withdrawal, please see our Privacy Policy .

Reviewer #1: **Yes: ** Alessandra Fierabracci

Reviewer #2: **Yes: ** Sammra Maqsood

---

## [Author Response · Author response to Decision Letter 1]

8 Mar 2025

Dear PLOS ONE Staff,

On behalf of all co-authors, I sincerely appreciate your constructive feedback and the opportunity to revise our manuscript titled "The Association between PTPN22 SNPs and Susceptibility to Type 1 Diabetes: An updated meta-analysis" (manuscript number: PONE-D-24-50318). We have carefully addressed each point raised in your review and detailed our responses below. Revised sections are marked in the manuscript, and supplementary files have been updated accordingly.

1.Tables 1 and 2 - Zak et al. 2023 examined patients from Armenia, not from America.

Response: We thank the editor for bringing this error to our attention. The geographic origin of the study population in Tables 1 and 2 has been corrected accordingly. Zak et al. (2023) is now accurately described as examining patients from Armenia.

2.Fig 2 - state clearly the reasons for exlusion of studies explicitly mentioned in the figure.

Response: We appreciate the suggestion to improve clarity. In the manuscript, we have clearly outlined the specific reasons for excluding certain studies. These reasons include studies with quality scores ≤12 and studies related to HWD. Additionally, to enhance transparency, we have provided a detailed explanation of the exclusion criteria in the Quality Assessment section of the manuscript

3.Fig. 1 - typo "titlis".

Response: We thank the editor for pointing out this typographical error. The word "titlis" in Figure 1 has been corrected to "titles." Additionally, we have thoroughly reviewed the manuscript and figures to identify and correct any other typographical errors, ensuring the accuracy of the content.

4.The whole study needs to better reflect the geographic gradient of the study polymorphisms within the Caucasian populations. There are enormous differences in the prevalence of the study rs in general and T1DM Caucasian populations both in the N-S and E-W directions throughout Europe. It is described the best for rs2476601. These differences are enormous and prevent generalization of conclusions for the whole Caucasian population.

Response: We thank the editors for their valuable comments regarding the need to consider geographic differences in the prevalence of the studied polymorphisms within Caucasian populations. In response, we have significantly revised the manuscript and Table 3 to address this concern. Specifically:

We excluded mixed-race populations and performed subgroup analyses based on geographic regions (Eastern, Western, Northern, Southern, and Central Europe). Our forest plots and heterogeneity analyses revealed notable regional variations in effect sizes. Heterogeneity was highest in Southern Europe (I² = 63.2%, p = 0.028) and moderate in Western Europe (I² = 59.3%, p = 0.086). In contrast, Eastern, Northern, and Central Europe exhibited minimal heterogeneity (I² = 0.0–10.5%, p > 0.3). The overall analysis demonstrated a significant correlation (z = 8.60, p < 0.001), consistent with findings from most regions. However, the results for Southern Europe were not statistically significant (z = 1.23, p = 0.217), potentially reflecting differences in population structure or allele frequencies.

These findings underscore the importance of accounting for geographic differences when interpreting genetic associations in Caucasian populations and further support the robustness of our conclusions for most regions. We have further discussed these differences and their implications in the revised discussion section of the manuscript.

5.I cannot open the supplementary files. Please double-check their integrity and reupload them.

Response: We apologize for the inconvenience caused by the issues with the supplementary files. The supplementary information file has been re-uploaded with the following adjustments: it has been converted to the Excel 97-2003 (.xls) format for broader compatibility, and its integrity has been verified using LibreOffice 25.2.1 and WPS Office. If the issue persists, we would be happy to provide the files in alternative formats (e.g., PDF or CSV).

Thank you for your detailed feedback and guidance. We hope that the revised manuscript meets the journal’s requirements. Please do not hesitate to contact us if further modifications are required. We look forward to your response.

Kind regards,

Fei-fei Wu

---

## [Editor Report · Decision Letter 2]

11 Mar 2025

The Association between PTPN22 SNPs and Susceptibility to Type 1 Diabetes: An updated meta-analysis

PONE-D-24-50318R2

Dear Dr. Wu,

We’re pleased to inform you that your manuscript has been judged scientifically suitable for publication and will be formally accepted for publication once it meets all outstanding technical requirements.

Kind regards,

Petr Heneberg

Academic Editor

PLOS ONE
---

## [Editor Report · Acceptance letter]

PONE-D-24-50318R2

PLOS ONE

Dear Dr. Wu,

I'm pleased to inform you that your manuscript has been deemed suitable for publication in PLOS ONE. Congratulations! Your manuscript is now being handed over to our production team.

Kind regards,

on behalf of

Dr. Petr Heneberg

Academic Editor

PLOS ONE